# Electrochemical Synthesis of 1,1′-Binaphthalene-2,2′-Diamines via Transition-Metal-Free Oxidative Homocoupling

Duona Fan, Md. Imrul Khalid ⓘ, Ganesh Tatya Kamble, Hiroaki Sasai and Shinobu Takizawa *

SANKEN, Osaka University, Suita Campus, Mihogaoka, Ibaraki-shi, Osaka 567-0047, Japan
* Correspondence: taki@sanken.osaka-u.ac.jp; Tel.: +81-6-6879-8467

**Abstract:** The facile and green synthesis of 1,1′-binaphthalene-2,2′-diamine (BINAM) derivatives was established via the anodic dehydrogenative homo-coupling of 2-naphthylamines. The sustainable protocol provided a series of BINAMs in excellent yields of up to 98% with good current efficiency (66%) and $H_2$ as the sole coproduct without utilizing transition-metal reagents or stoichiometric oxidants.

**Keywords:** electrochemical synthesis; 2-naphthylamines; anodic dehydrogenative homocoupling; BINAMs

## 1. Introduction

1,1′-Bi-2-naphthylamine (BINAM) and its derivatives are widely used as building blocks for transition-metal ligands and organocatalysts [1–4], as well as chiroptical materials for fluorescence sensing [5–7]. Among their syntheses, transition-metal mediated coupling of 2-naphthylamine derivatives has been well established [8–12]. In particular, the Ullmann [13] and metal-mediated C($sp^2$)–H oxidative [14–20] coupling reactions of 2-naphthylamines are the most popular synthetic approaches to obtaining BINAMs. Benzidine rearrangement [21–23] and Smiles rearrangement of phenolic compounds [24] have been reported to be alternative synthetic methods for BINAMs. However, the regio- and chemoselective C–C coupling of 2-naphthylamines remains challenging, probably because these strategies require the use of excess amounts of metals and oxidants, which leads to many side reactions of starting material and over-oxidation of coupling products.

Biaryls synthesis through anodic oxidation utilizing electrochemical synthesis has emerged as a promising green and sustainable approach over the last few decades. The electrochemical dehydrogenative coupling of aryls utilizes electricity as an alternative to oxidants and produces $H_2$ as the sole coproduct without generating any toxic waste; thus, this approach exhibits great advantages in terms of high atom economy and environmentally benign synthesis protocols [25,26]. Although remarkable achievements have been made in the electrochemical dehydrogenative heterocoupling of anilines [27], few reports on the homocoupling of aniline derivatives [28,29], particularly 2-naphthylamines, are available. In the 1980s, Gossage [30] and Sereno [31] independently described the electrochemical dehydrogenative homocoupling of 2-naphthylamines, which afforded BINAMs, but in low yield. Thus, the development of efficient oxidative homocoupling protocols for 2-naphthylamines via electrochemical synthesis is of great importance.

Because the discharge of the electrolyte or solvent leads to a low yield and poor current efficiency, we reexamined the anodic dehydrogenative homocoupling conditions of 2-naphthylamines to save energy and chemical loading in the pursuit of developing environmentally benign chemical reactions. To our delight, under the newly established conditions, the corresponding homocoupling products were obtained in 98% yield with good current efficiency (66%; Scheme 1).

**Scheme 1.** Electrochemical dehydrogenative homocoupling of 2-naphthylamines.

## 2. Materials and Methods

### 2.1. Materials

*N*-Phenyl-2-napthylamine (**1a**) was purchased from Tokyo Chemical Industries (TCI). All commercially available organic and inorganic compounds were used directly without further purification.

### 2.2. Methods

#### 2.2.1. Spectroscopy and Spectrometry

$^{1}$H- and $^{13}$C-NMR spectra were recorded at 25 °C using a JEOL JMN ECS400 FT NMR instrument ($^{1}$H-NMR 400 MHz; $^{13}$C-NMR 100 MHz). The $^{1}$H-NMR spectra are reported as follows: chemical shift in ppm downfield of tetramethylsilane and referenced to a residual solvent peak (CHCl$_3$) at 7.26 ppm, integration, multiplicities (s = singlet, d = doublet, t = triplet, q = quartet, m = multiplet), and coupling constants (Hz). The $^{13}$C-NMR spectra are reported in ppm relative to the central line of the triplet for CDCl$_3$ at 77.16 ppm. ESI-MS spectra were obtained using a JMS-T100LC instrument (JEOL). FT-IR spectra were recorded using a JASCO FT-IR system (FT/IR4100). Thin-layer chromatography (TLC) analysis of the reaction mixture was performed on Merck silica gel 60 F254 TLC plates and visualized under UV light. Column chromatography on SiO$_2$ was performed using Kanto Silica Gel 60 (63–210 μm).

#### 2.2.2. General Protocol for the Anodic Homocoupling of 2-Naphthylamines

ElectraSyn 2.0 and platinum were utilized as the reaction device and electrode, respectively. A suspension of 2-naphthylamines (0.1 mmol) and $^{n}$Bu$_4$NPF$_6$ (0.1 M) in 1,1,1,3,3,3-hexafluoro-2-propanol (HFIP) (5 mL) was added to an undivided vessel and stirred under a constant current of 4 mA for 2 h. The electrolyte was removed using short silica gel column chromatography ($^{n}$hexane/ethyl acetate = 1/1). The fraction was dried, and the crude product was purified by silica-gel column chromatography ($^{n}$hexane/ethyl acetate = 20/1) to afford the pure homocoupling product.

## 3. Results

### 3.1. Optimization of the Reaction Conditions

Initially, we screened the electrodes, solvent systems, currents, and electrolytes to determine the optimal conditions (Table 1, also see supporting information). The electrodes were screened by employing 0.1 mmol *N*-phenyl-2-naphthylamine (**1a**, oxidative potential 1.05 eV; see Supporting Information) as the model substrate. The platinum electrode exhibited good reactivity, affording homocoupling product **2a** in 98% yield (current efficiency, 66%) without the formation of any side product (entry 1). In contrast, the carbon–platinum [32] and fluorine-doped tin oxide(FTO) [33] electrodes gave **2a** in 43% and 29% yields, respectively, with low current efficiencies (entries 2 and 3). Other alcoholic reaction solvents, such as methanol, ethanol, and trifluoroethanol, reduced the yield of **2a** to 6–12% with current efficiencies of 4–8% (entries 4–6), along with a 5% yield of aza [5]

helicene. HFIP, an appropriate solvent for the reaction, serves as an excellent hydrogen bond donor and provides highly persistent radical cations [34,35]. The effect of a constant current was also investigated. As shown in entry 7, when a current of 2 mA was employed for the electrosynthesis process, the current efficiency increased to 80%, but the yield of **2a** decreased to 60%. In contrast, employing a current of 6 mA for the electrosynthesis process led to the formation of **2a** in 49% yield, with a current efficiency of 22% (entry 8). Suppressing the discharge of the electrolyte or solvent resulted in higher yield and current efficiency. Among the electrolytes we screened, $^{n}Bu_4NPF_6$ proved to be superior to $LiClO_4$ (**2a**, 57% yield due to low solubility in HFIP) and $^{n}Bu_4NClO_4$ (**2a**, 29% yield) (entries 9 and 10). No reaction occurred in the absence of electricity (entry 11).

**Table 1.** Optimization of the conditions for the electrochemical homocoupling of 2-naphthylamines using **1a** as the model substrate.

| Entry | Variation from Standard Conditions | % Yield (% Current Efficiency) [a] | |
|---|---|---|---|
| | | **2a** | **3a** |
| 1 | None | 99, 98 [b] (66) | - |
| 2 | C (+)/Pt (−) | 43 (29) | - |
| 3 | FTO (−)/FTO [c] (+) | 29 (20) | - |
| 4 | MeOH instead of HFIP | 10 (7) | 5 |
| 5 | EtOH instead of HFIP | 6 (4) | 5 |
| 6 | $CF_3CH_2OH$ instead of HFIP | 12 (8) | - |
| 7 | 2 mA | 60 (80) | - |
| 8 | 6 mA | 49 (22) | - |
| 9 | with $LiClO_4$ instead of $^{n}Bu_4NPF_6$ | 57 [d] (38) | - |
| 10 | with $^{n}Bu_4NClO_4$ instead of $^{n}Bu_4NPF_6$ | 29 (19) | - |
| 11 | No electricity | No reaction | |

[a] NMR yield; [b] Isolated yield; [c] FTO = Fluorine-doped tin oxide; [d] HFIP:MeOH (3:2) for 8 h.

### 3.2. Scope of Substrates

With the optimized conditions in hand (reaction solvent: HFIP, electrolyte: $^{n}Bu_4NPF_6$ (0.1 M), electrode: platinum, constant current: 4 mA, and reaction temperature: 25 °C), we investigated the substrate scope of the 2-naphthylamines (Figure 1). The electrochemical homocoupling of 2-naphthylamines **1** proceeded smoothly at moderate current efficiencies (44–66%). *N*-Phenyl-2-naphthylamine (**1a**) produced homocoupling product **2a** in 98% yield. *N*-4-Tolyl and *N*-2-tolyl-2-naphthylamines (**1b** and **1c**), respectively, were also found to be appropriate coupling precursors, giving the corresponding homocoupling products **2b** and **2c** in 92% and 83% yields, respectively. The reaction of substrates **1d** and **1e** with electron-donating or electron-withdrawing groups, such as *N*-2,3-dimethoxyphenyl and *N*-4-bromophenyl, on the nitrogen atom, showed good functional group tolerance, giving binaphthylamines **2d** and **2e** in good yields. When *N*-(1,1′-biphenyl)-4-yl-2-naphthylamine (**1f**) was used as the substrate, a moderate yield of the coupling product **2f** was obtained because of the poor solubility of **1f** in HFIP. The reactions of *N*-alkyl-2-naphthylamines, such as *N*-methyle-2-naphthylamine (**1g**), *N*-ethyl-2-naphthylamine (**1h**), *N*-isopropyl-2-naphthylamine (**1i**), and *N*-*t*-butyle-2-naphthylamine (**1j**) afforded the corresponding homocoupling products **2g**, **2h**, **2i** and **2j** in 30%, 87%, 85% and 72% yields, respectively. The dehydrogenative coupling reactions of *N*-aryl-2-naphthylamines (**1k–1n**) with various substituents were also conducted. Products **2k** and **2l** with methoxy groups **2m** and **2n** with phenyl and cyano groups were readily obtained in good to excellent yields (65–95%

yields). *N*-Naphthyl-2-naphthyl amine (**1o**) could not be tolerated. Finally, *N*-benzyl-naphthylamine (**1p**) was employed in the electrochemical homocoupling reaction. The corresponding product **2p** was obtained an 85% yield and could be transformed into BINAM with Lewis acid (See Supporting Information).

*a* Reaction conditions: 2-naphthylamine **1** (0.1 mmol, 1.0 equiv.) and *n*Bu₄NPF₆ (0.1 M) in HFIP (5 mL) were added to an undivided vessel and stirred under a constant current of 4 mA for 2 h; *b* Isolated yield; *c* Current efficiency in parentheses. *d* 0.4 mmol of **1a**.

**Figure 1.** Substrate scope of *N*-naphthylamines *a–d*.

## 4. Discussion

In our study, the homocoupling reaction of 2-naphthylamines proceeded smoothly with good current efficiency. To further understand the relevant mechanism, we conducted the heterocoupling reactions of **1b** with *N*-4-tolyl (an electron-donating group) and **1e** with *N*-4-bromo-phenyl (an electron-withdrawing group) under the optimal conditions (Scheme 2).

Homocoupling product **2b** was obtained in 90% yield as the major product, along with homocoupling product **2e** in 10% yield and trace amounts of heterocoupling product **2be**. The relative proportions of the products arising from the radical–radical coupling reactions are aligned with their relative reactivities [36–38]. In principle, **1e** is less oxidizable than **1b**, which should result in the formation of homocoupling **2b** as the major product. The results support our hypothesis that the present coupling reaction of **1** proceeds through radical–radical coupling, as shown in Scheme 3. Triggering by single-electron transfer (SET) of **1a** on the anode made the formation of intermediate **I**. The generated **I** species could be

in equilibrium into **Ia** and **Ib** by electron transfer. Then a radical–radical coupling of **Ib** proceeded to afford the coupling product **2b** through the oxidation of intermediate **II**. On the cathode, the generated H$^+$ was reduced to give H$_2$ as a sole coproduct.

**Scheme 2.** Electrochemical heterocoupling of substrate **1b** and **1e**.

**Scheme 3.** Proposed mechanism for the homocoupling of 2-naphthylamines.

## 5. Conclusions

We developed a facile and sustainable protocol for the homocoupling of various 2-naphthylamines with up to 98% yield and good current efficiency (66%). This new protocol not only saves energy and chemical loading but also significantly improves the product yield, thus representing a significant improvement to the previous electrosynthesis approach. Investigations of the further applications of homocoupling products are ongoing in our laboratory.

**Supplementary Materials:** The supporting information can be downloaded at: https://www.mdpi.com/article/10.3390/suschem3040034/s1; Figure S1: IKA device ElectraSyn 2.0 standard setup; Table S1. Screening solvent of constant current for optimizing reaction conditions; Figure S2: CV experiments (MeCN) as a solvent with Bu$_4$NPF$_6$ (0.1 M) as an electrolyte.

**Author Contributions:** S.T. and H.S.: conceptualization, resources, writing—reviewing, and editing. D.F.: investigation, resources, visualization, validation, writing—original draft. M.I.K., and G.T.K.: investigation, resources, and validation. All authors have read and agreed to the published version of the manuscript.

**Funding:** This work was supported by JSPS KAKENHI Grant Numbers 22K06502 in Grant-in-Aid for Scientific Research (C), Transformative Research Areas (A) 21A204 Digitalization-driven Transformative Organic Synthesis (DigiTOS), 22KK0073 in Fund for the Promotion of Joint International Research (Fostering Joint International Research (B)) from the Ministry of Education, Culture, Sports, Science, and Technology (MEXT), and the Japan Society for the Promotion of Science (JSPS), JST CREST (No. JPMJCR20R1), and Hoansha Foundation.

**Institutional Review Board Statement:** Not applicable.

**Informed Consent Statement:** Not applicable.

**Data Availability Statement:** Data available in a publicly accessible repository.

**Acknowledgments:** We acknowledge the technical staff of the Comprehensive Analysis Center of SANKEN, Osaka University (Japan).

**Conflicts of Interest:** The authors declare no conflict of interest.

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
