# Peer review of "Electrochemical Synthesis of 1,1′-Binaphthalene-2,2′-Diamines via Transition-Metal-Free Oxidative Homocoupling"

_2673-4079, doi:10.3390/suschem3040034_

Round 1

Reviewer 1 Report

This manuscript by Takizawa and co-workers describes the anodic C-C cross-coupling of 2-naphthylamines, where the radical cation species were coupled each other to give the corresponding BINAM derivatives (12 substrates in 66-98% yields). The heterocoupling between 1b and 1e resulted in the preferential formation of 2b, suggesting the stability of radical cation species plays significant role in the current C-C cross coupling chemistry. Given that the current results certanly updated the previous chemistry from 1980's (ref 30 and 31), it is the opinion of this reviewer that the current manuscript presented a sustainable synthetic method that possesses further application potential in the BINAM syntheses. Based on the above assessment, this reviewer strongly supports the immediate publication of the present manuscript in Sustainable Chemistry. 

The Supporting Information is thoroughly prepared, and the manuscript is well written. Only one typo in Table 2 legend---0.1 mmol (space needed between number and mmol). 

Author Response

Reviewer 1: Comments and Suggestions for Authors

This manuscript by Takizawa and co-workers describes the anodic C-C cross-coupling of 2-naphthylamines, where the radical cation species were coupled each other to give the corresponding BINAM derivatives (12 substrates in 66-98% yields). The heterocoupling between 1b and 1e resulted in the preferential formation of 2b, suggesting the stability of radical cation species plays significant role in the current C-C cross coupling chemistry. Given that the current results certanly updated the previous chemistry from 1980's (ref 30 and 31), it is the opinion of this reviewer that the current manuscript presented a sustainable synthetic method that possesses further application potential in the BINAM syntheses. Based on the above assessment, this reviewer strongly supports the immediate publication of the present manuscript in Sustainable Chemistry. 

Reviewer 1-1: The Supporting Information is thoroughly prepared, and the manuscript is well written. Only one typo in Table 2 legend---0.1 mmol (space needed between number and mmol). 

Our response: We appreciate your comments. We revised these parts accordingly.

Reviewer 2 Report

The article describes an improved electrosynthetic procedure for the synthesis of binapthalene-diamine derivatives with respect to previous reports (refs 30 and 31).

The results are indeed remarkable in terms of reactions conditions and yields. The authors also report one experiment under competitive conditions with the aim to confirm the expected mechanism. The reaction conditions are well described in the SI which also contain the procedure for the synthesis of the substrates and product characterization.

The article is publishable in Sustainable Chemistry as it fits well in the scope of the journal. However, the following points should be taken into consideration in the preparation of the revised article.

The authors should attempt to scale up the reaction at least in one or two cases as 0.1 mmol of substrate is indeed a low amount from a synthetic point of view.

Detailed experimental procedure (including mg of substrate and products) and proofs of the results of the competition experiment between 1b and 1e should also be given. If the relative yields of products were determined by analysis of the crude reaction mixture, figures of the corresponding NMR spectra or HPLC chromatogram should be reported. Regarding to the result, it appears odd that the omocoupled product 2e (derived from the least reactive substrate) is in larger amount than the cross coupled product 2be.

Page 2 line 68: the acronym used for the solvent HFIP should be expressed in full, as the solvent is such a critical parameter.

Page 3 line 99:…. isolated yield…

Author Response

Reviewer 2: Comments and Suggestions for Authors

The article describes an improved electrosynthetic procedure for the synthesis of binapthalene-diamine derivatives with respect to previous reports (refs 30 and 31). The results are indeed remarkable in terms of reactions conditions and yields. The authors also report one experiment under competitive conditions with the aim to confirm the expected mechanism. The reaction conditions are well described in the SI which also contain the procedure for the synthesis of the substrates and product characterization. The article is publishable in Sustainable Chemistry as it fits well in the scope of the journal. However, the following points should be taken into consideration in the preparation of the revised article.

Reviewer 2-1: The authors should attempt to scale up the reaction at least in one or two cases as 0.1 mmol of substrate is indeed a low amount from a synthetic point of view.

Our response: We carried out 0.4 mmol-scale reaction of 1a. The reaction produced the homocoupling product 2a in 98% isolated yield, which is comparable to a 0.1 mmol-scale (see Table 2).

Reviewer 2-2: Detailed experimental procedure (including mg of substrate and products) and proofs of the results of the competition experiment between 1b and 1e should also be given. If the relative yields of products were determined by analysis of the crude reaction mixture, figures of the corresponding NMR spectra or HPLC chromatogram should be reported.

Our response: We described the experimental procedure the heterocoupling reaction of 1b and 1e in supporting information. The determination of the each ration of products by using the NMR and HPLC analysis was difficult because of overlapping of each peak, finally, the crude product was purified by silica-gel column chromatography (n-hexane/ethyl acetate = 20/1) to afford 2b in 90%, 2e in 10%, trace amount of 2be, along with recovery of 1a and 1e.

Reviewer 2-3: Regarding to the result, it appears odd that the omocoupled product 2e (derived from the least reactive substrate) is in larger amount than the cross coupled product 2be.

Our response: The relative proportions of the coupling products arising through the radical-radical coupling reactions are aligned with their relative reactivities. In principle, 1e is less oxidizable, which should result in the formation of the homo-coupling 2b as the major product. On the time course study (reaction time: 2 hours), 1b was initially consumed and then homo-coupling of 1e gradually began. The prolonging reaction time led to full conversion of 1b and 1e to give the corresponding coupling products in high yields. Additional investigations into the reaction mechanism is in process in our laboratory.

Reviewer 2-4: Page 2 line 68: the acronym used for the solvent HFIP should be expressed in full, as the solvent is such a critical parameter.

Response: It has been corrected and refined accordingly.

Reviewer 2-5: Page 3 line 99:…. isolated yield…

Response: It has been corrected and refined accordingly.

Reviewer 3 Report

This paper reported a simple and green synthesis of 1,1 ′ -double naphthalene-2,2 ′ -diamine (BINAM) derivatives by anodic dehydrogenation coupling of 2-naphthalamide. The protocol provides a range of excellent yields and H2 as the only byproduct, without using transition metal reagents or stoichiometric oxidants. It is a meaningful paper that contains interesting results worth to be published. Some modifications are required f for the benefit of the reader as follows:

1. The optimization conditions are not detailed enough, it can be seen that only two current efficiency are given in the Table 1. The result is not very ideal. More electrochemical details such as current efficiency needs to be given in the manuscript.

2. The substrate expansion part it too limited. Will you please give more examples.

3. The reaction mechanism diagram should be explained. The discussion is not fully reflected in the reaction mechanism.

Author Response

Reviewer 3: Comments and Suggestions for Authors

This paper reported a simple and green synthesis of 1,1 ′ -double naphthalene-2,2 ′ -diamine (BINAM) derivatives by anodic dehydrogenation coupling of 2-naphthalamide. The protocol provides a range of excellent yields and H2 as the only byproduct, without using transition metal reagents or stoichiometric oxidants. It is a meaningful paper that contains interesting results worth to be published. Some modifications are required f for the benefit of the reader as follows:

Reviewer 3-1: The optimization conditions are not detailed enough, it can be seen that only two current efficiency are given in the Table 1. The result is not very ideal. More electrochemical details such as current efficiency needs to be given in the manuscript.

Our response: We appreciate your important comments. We carefully recheck the reaction conditions including current efficiency (see supporting information, Table S1).

Reviewer 3-2: The substrate expansion part it too limited. Will you please give more examples.

Our response: Four substrates 1g, 1j, 1m, and 1o were carried out accordingly (see Table 2 in manuscript).

Reviewer 3-3: The reaction mechanism diagram should be explained. The discussion is not fully reflected in the reaction mechanism.

Our response: We inserted the following sentences for the mechanism explanation: “Triggering by single-electron transfer (SET) of 1a on the anode made the formation of intermediate I. The generated I species could be in equilibrium into Ia and Ib by electron transfer. Then a radical-radical coupling of Ib proceeded to afford the coupling product 2b through the oxidation of intermediate II. On the cathode, the generated H+ was reduced to give H2 as a sole coproduct.

Reviewer 4 Report

In this manuscript the Authors report the coupling of di-arylamines by exploiting an electrochemical 2e-oxidation of arylnaphthylamines. Although very similar approaches have already been reported by others (as indicated by the references), better yields and higher current efficiencies have been achieved in this work. The research has been well designed and references as well as product characterization are appropriate. I believe that this work deserves publication providing that the following remarks are addressed: 1) In Table 1 is missing the note referring to the yield in brackets (current efficiency). 2) Most importantly, the Authors should clearly indicate how they prevented the corrosion of the standard platinum electrode used as an anode. Indeed, these electrodes (Pt coated on copper) are not suitable for preparative anodic electrolysis: this could really hamper their use in large scale electrosynthesis.

Author Response

Reviewer 4: Comments and Suggestions for Authors

In this manuscript the Authors report the coupling of di-arylamines by exploiting an electrochemical 2e-oxidation of arylnaphthylamines. Although very similar approaches have already been reported by others (as indicated by the references), better yields and higher current efficiencies have been achieved in this work. The research has been well designed and references as well as product characterization are appropriate. I believe that this work deserves publication providing that the following remarks are addressed:

Reviewer 4-1: In Table 1 is missing the note referring to the yield in brackets (current efficiency).

Our response: We appreciate your comments. We revised Table 1 involving the foot note.

Reviewer 4-2: Most importantly, the Authors should clearly indicate how they prevented the corrosion of the standard platinum electrode used as an anode. Indeed, these electrodes (Pt coated on copper) are not suitable for preparative anodic electrolysis: this could really hamper their use in large scale electrosynthesis.

Our response: We appreciate your comments, and also agree your suggestion. However, during our investigation, few corrosions of the platinum anode (not Pt coated on copper) was observed. Toward the large-scale synthesis, their flow electrochemical synthesis with appropriate and durable electrodes are under in progress.

Round 2

Reviewer 3 Report

This manuscript is acceptable after the major revision.